# Today Is My Day: Analysis of the Awareness Campaigns’ Impact on Functional Diversity in the Press, on Google, and on Twitter

**DOI:** 10.3390/ijerph18157789

**Published:** 2021-07-22

**Authors:** Irene Gómez-Marí, Pilar Sanz-Cervera, Raúl Tárraga-Mínguez

**Affiliations:** Department of Education and School Management, Faculty of Teacher Training, University of Valencia, 46022 Valencia, Spain; pilar.sanz-cervera@uv.es (P.S.-C.); raul.tarraga@uv.es (R.T.-M.)

**Keywords:** functional diversity, Google, awareness campaign, media, press, Twitter

## Abstract

(1) Every day, people with functional diversity face different kinds of difficulties that pose a barrier to their social inclusion. These difficulties often go unnoticed by most citizens. Social networks are a powerful tool to sensitize the population. With this objective, different organizations such as associations, federations, foundations, and other institutions have promoted campaigns through the celebration of world days for different types of functional diversity. This research aims to monitor and analyze the impact of these social campaigns in Spain, including Asperger’s syndrome, rare diseases, Down syndrome, autism, hearing and visual impairment, cerebral palsy, dyslexia, ADHD, spina bifida, disability, and dyscalculia world days, between 2015 and 2020. (2) The impact of each campaign on the press, Google, and Twitter has been analyzed using: MyNews, Google Trends, and Trendinalia. (3) The results suggest a close relationship between the impact on the number of pieces of news generated in the press, the searches on Google, and the hashtags in high positions on Twitter. (4) The campaigns with the greatest levels of success are those whose diagnoses involve greater difficulties in adaptive behavior. These results can provide some practical implications for future campaigns.

## 1. Introduction

The celebration of world days was born with the aim of raising awareness about topics related to issues of interest, such as human rights or health [1]. One of the mechanisms that the promoters of the campaigns employ to achieve this goal is to attract the attention of the media, so that they become a speaker denouncing situations that make the full inclusion of people with functional diversity into the public difficult, as well as to pressure governments to take action to address these problems [1,2,3]. People with functional diversity are a vulnerable group at risk of exclusion in many contexts [4]. Numerous barriers related to accessibility, lack of resources, and economic support still exist, as does the existence of stigmas of different types, affecting the socialization processes of people with functional diversity [4,5]. In recent decades, different associations, federations, and national and international institutions have created awareness days for different diagnoses [6,7], and they have turned the celebration of these days into a tool to try to break down barriers related to prejudice, stigma, ignorance, and discrimination, as well as to promote the dissemination of quality information that raises awareness of different realities in relation to normotypic ones [6,7,8,9].

### 1.1. Awareness Campaigns for Functional Diversity

We will consider as awareness campaigns all those activities whose motivation is the dissemination of a common message related to a diagnosis or a disease in order to raise awareness about a specific condition. Those tend to use advertising in the media to promote any of the subjects involved in the cause defended. The awareness campaigns that are carried out on the occasion of the celebration of world days can promote the fight against stigmatization, misconceptions, prejudice, and ignorance [6]. Campaigns for these days are organized by associations, federations, foundations, and national, international, and global institutions that promote the spreading of quality information and raise awareness about some hidden realities [8]. Thus, those in charge of designing and implementing these awareness-raising campaigns tend to influence public attitudes and beliefs, creating and disseminating contents to help to shape ideas embedded in the social imagination [6]. To achieve this, it is necessary to reach the greatest number of target audiences; therefore, campaigns are often publicized through channels that allow for the maximizing of potential target audiences, such as traditional media or social networks.

However, this dissemination can be positive or negative, depending on the content spread for some campaigns. A recent study [10] concluded that, in some cases, the publication of pieces of news about a diagnosis can be controversial. For instance, the Greta Thunberg phenomenon can help to better understand autism or Asperger’s syndrome, but it can also stigmatize these syndromes by relating them to this particular character. Moreover, in some cases, campaigns do not manage to influence the media.

What is fairly apparent is that traditional media, such as the press, is able to configure the agenda, determining what are topics of interest for citizens; the press is thus a powerful influence on the shaping of public opinion [11]. Social networks allow campaigns to reach more users, while other mechanisms available via the Internet, such as websites, help to spread the true face of diagnoses [2,8,9,12,13,14,15].

### 1.2. Functional Diversity

The term “functional diversity” was born with the aim of overcoming the old concept of “disability” [16]. This term is a condition in which the existence of “different capacities” is defended, leaving behind the conceptions of “deficiencies”, “limitations”, “restrictions”, and “handicaps”, all of which place the person with functional diversity in a position of inaction. This new conception considers that everybody has different ways of functioning, i.e., different ways of carrying out everyday tasks such as walking, reading, communicating, cooking, speaking, etc. However, the indistinct use of “disability” and “functional diversity” still persists today. Even some of the awareness-raising campaigns analyzed in the present study bear “disability” in their names. That is why, throughout this study, we refer to both nomenclatures depending on context: when we refer to campaigns whose names are already established using “disability”, we will talk about “disability”; regarding the presentation of our results, discussions, and conclusions, we will refer to “functional diversity”, as this is a more inclusive term that includes the different needs and realities of people with any diagnoses.

In this study, we analyze the impact of the world days campaigns for twelve types of diversity, following the prevalence data in Spain for the different types of functional diversity that are expressed in a Spanish report [17]. It concluded that, in Spain, there are 217,416 students with special educational needs being trained in mainstream schools. Of them, 3.9% have hearing impairments; 6.3% have a motor disability; 31.2% have an intellectual disability; 1.6% are visually impaired; 15.5% have a neurodevelopmental disorder; 22.1% have a behavior disorder; and 4.7% have comorbid impairments. In total, 79.1% of the students with functional diversity are taught in primary and secondary stages. For this reason, we chose these kinds of functional diversity. 

Table 1 summarizes all of these forms of functional diversity and provides some explanation about each one:

### 1.3. Previous Studies on the Impact of Awareness Campaigns on Society

Prior studies carried out to assess the effectiveness of these kind of campaigns have used different strategies [1,2,4,6,9,13,15,18,19,20,21,22,23,24,25,26]. Some of them have focused their efforts on analyzing the presence of content related to campaigns in the media, particularly in the press. Specifically, there are studies that have analyzed the frequency and treatment received in press of campaigns on diagnoses such as autism [13,22], Down syndrome [23], rare diseases [24,25], or functional diversity in general [26].

Some prior studies have focused on analyzing whether campaigns coincide with an increase in Internet searches for content related to the campaigns themselves. In particular, they have analyzed Google’s search patterns, as Google accounts for almost 80% of Internet searches [15]. Thus, campaigns on diagnoses such as malaria [9], sepsis [1], thrombosis [2], or autism [18] have used Google to measure the impact of their respective campaigns. Other searches have focused on assessing the impact of campaigns on Twitter, a social network that, while not exempt from being involved in many problems related to digital content, has become a platform that can contribute to the fight against disinformation and misconceptions [19]; Twitter can also increase the visibility of different diagnoses [20]. This is the case in studies that have evaluated the effectiveness of visual disability campaigns [21] or World Autism Awareness Day [6].

Although numerous studies assess the impact of these awareness-raising campaigns on searches in Google, social networks, or media, thus far, there are only a few studies that have assessed the impact on several media simultaneously (at least two). In this sense, [15] carried out an analysis of media and assessed the effectiveness of a World Donor Day Campaign through the analysis of Google searches and the increase in the number of likes on Facebook sites; [4] evaluated the impact of an Epilepsy Awareness Campaign according to Twitter hashtags and the volume of Google searches. However, these kinds of analysis of the subject, i.e., combining several media, are unusual in the scientific literature.

### 1.4. Purpose of the Study

The main goal of this study is to analyze the effectiveness of Spanish campaigns regarding 12 awareness campaigns related to functional diversity during the period from 2015 to 2020. This study attends to the following: (a) the number of pieces of news published in press related to each specific campaign; (b) the number of Google searches related to the campaign’s content; and (c) the number of trending topics generated on Twitter during the campaign. The awareness campaigns analyzed (sorted by date) are as follows: Asperger syndrome, rare diseases, Down syndrome, autism, hearing impairment, visual impairment, cerebral palsy, dyslexia, ADHD, spina bifida, disability (understood as a generic term that includes people with functional diversity), and dyscalculia.

## 2. Methods

The platforms chosen to carry out our study were press (print and digital), Google, and Twitter. In the case of press analysis, a lot of previous research has investigated the image of functional diversity in the press, acting as traditional mass media [13,22,23,24,25,26]; however, there is no considerable previous research regarding this issue in Spain. Therefore, we intended to analyze it. In the case of Google, some research [15] contended that it is the search tool that accounts for almost 80% of Internet searches. In European countries, more than 90% of Internet users seek information on Google [https://netmarketshare.com/search (accessed on 17 December 2020)]. Therefore, Google creates an important database for carrying out research. Additionally, we eventually decided to assess these campaigns’ impacts on Twitter because, according to https://de.statista.com/ (accessed on 17 December 2020), it is a social media platform with about 353 million monthly active users and a high level of dissemination. For these reasons, we considered it essential to understand how campaigns have had an impact on this platform.

### 2.1. Instruments

Three tools were used to gather all data included in the current study: MyNews, Google Trends, and Trendinalia.

MyNews is a digital newspaper library that stores information from more than 1500 Spanish media from 1996 onwards. MyNews allows users to locate pieces of news for a given term on a date or a particular period of time. The tool has filters to narrow the searches to international newspapers, national, regional, or local news, as well as digital or printed news; these filters specify whether a keyword appears in any part of the piece of news or only in the headline.Google Trends is an online and open database that allows one to analyze the relative search volume of queries conducted in the Google search engine at any time or place. Therefore, Google Trends does not inform one of the results that Google offers, but rather allows one to infer the degree of interest among Internet users that a certain topic arouses at any given time or in a certain region. It provides a relative popularity value, where 100 represents the moment of greatest searches for a period, place, and term chosen, and 0 represents complete disinterest. Therefore, this tool has allowed us to determine the moments in each of the analyzed years in which Google users have shown more interest in the search terms related to the analyzed diagnoses in this article.Trendinalia is a tool that offers daily trending topic (TT) lists on Twitter for a specific date and geographical region, providing information on the most popular hashtags, as ordered by the number of hours and minutes they have remained as subjects of maximum interest on a specific date.

### 2.2. Procedure

First, an initial search was conducted on the United Nations website [https://www.un.org/es/sections/observances/international-days/index.html (accessed on 17 December 2020)] and on the “International Day Of” site [https://www.diainternacionalde.com/busqueda (accessed on 17 December 2020)], where international days are collected; this search was performed with the aim of specifying the dates of the campaigns for the functional diversity conditions included in the present study. Additionally, to confirm this information, as well as the true implementation of these dates in Spain, a search was conducted on the websites of the main national associations regarding the selected diagnoses in order to corroborate or decide when the campaign was to be held in Spain, and therefore to provide a date as to when it might cause some reaction in Spanish society. After this double check, the dates considered as the institutional diagnosis days were added to Table 2.

Data collection procedure in MyNews. In order to know the number news pieces published at a national level during the week of the celebration of each campaign, we conducted independent searches for all years between 2015 and 2020 on MyNews. Searches were enacted in the 23 most widely read national newspapers according to the Association for the Research of Media (AIMC), excluding exclusively international media, regional, or local media. In total, according to the AIMC, and applying our screening criteria, the selected ones were as follows: 20 min, 20 minutos.es, ABC, abc.es, As, as.com, El Mundo, elmundo.es, El País, elpais.com, El Periódico, elperiodico.com, eldiario.es, elespanol.com, La Razón, larazon.es, La Vanguardia, lavanguardia.com, Marca, marca.com, okdiario.com, Público, and publico.es. A total of 25 searches with 25 keywords for all the campaigns for every single newspaper and in each of the years, between 2015 and 2020, were carried out. After that, a qualitative review of the headlines was conducted in order to eliminate the pieces of news in which the keywords appeared, but whose content was not related to the diagnosis. We then counted the pieces of news that resulted from the search with the indicated filters. The pieces of news whose headline was repeated several times were only counted once.Data collection procedure in Google Trends. To analyze the search patterns on Google regarding the different diagnoses studied, we performed independent searches in Google Trends using each of the terms listed in Table 2 for each year from 2015 to 2020, and we limited the results to searches carried out in Spain. A total of 72 searches were therefore carried out (corresponding to 6 searches for each of the 12 analyzed diagnoses in the current paper). The noted value corresponds to the search volume of the week in which each campaign was framed.Data collection procedure in Trendinalia. To locate the topics that became TT in Spain during the days of the analyzed campaigns, a search was conducted in Trendinalia for each national or international day in the covered period between 2015 and 2020, and we limited the results to Spain. Therefore, a total of 72 searches were conducted. All the TT of each search were tracked to identify, through a qualitative analysis, possible hashtags that bore some relation to the diagnoses studied. To achieve this, we not only took into account the terms of the diagnoses themselves, but also resorted to the websites of the associations to locate the possible hashtag they had prepared and promoted during the campaigns. The data found in Trendinalia were the number of TT related to each diagnosis, the position they were ranked, and the number of hours remaining as TT among the most viral hashtags on Twitter.

## 3. Results

Table 3 shows the results of the searches for the awareness campaigns included in this article during the period between 2015 and 2020. The included data in the table are the number of pieces of news related to the respective diagnoses that were published in the selected media (the total pieces of news from 2015 to 2020), the relative value provided by Google Trends regarding searches about those diagnoses (specifically, the arithmetic mean for the period 2015–2020), the number of TT identified by Trendinalia related to the campaign, and the time during which the identified hashtags remained in a position of TT (in both cases, the sum of the period 2015–2020). The values related to the pieces of news and Google searches refer to the full week of the studied awareness day. The amount and time of permanence of hashtags in TT positions refer only to the date of the day of the campaign.

Table 4, Table 5 and Table 6 describe more accurately the campaigns for each year, according to the impact achieved in the three studied fields: press, Google, and Twitter. Dyslexia and dyscalculia campaigns are excluded from Table 4, Table 5 and Table 6 as there is no impact on any of the dates in at least one of the analyzed fields.

### 3.1. Impact on the Press

Rare diseases, Down syndrome, autism, and disability are the campaigns that have the greatest impact on the generated pieces of news, with each one reaching over 200 items. Especially noteworthy is the campaign of Día Internacional de las Personas con Discapacidad, which reached 278 pieces of news during the analyzed period. The Asperger’s syndrome campaign attained a total of 50 pieces of news throughout the celebrated campaigns during the studied period. The rest of the campaigns did not exceed 32 pieces of news (which implies five pieces of news a year in the entire set of newspapers included in this search).

### 3.2. Impact on Google

Of the twelve awareness campaigns included, there are four cases in which the week with more volume of searches on Google for all the reviewed years coincides with the diagnosis awareness campaign. This is the case for Down syndrome, rare diseases, autism, and disability campaigns. The other six campaigns included in Table 3 present more variable search patterns, with values between 65.5 (Día Mundial de la Vista) and 90.3 (Día Internacional de las Personas Sordas). Five of these diagnoses (hearing impairment, visual impairment, ADHD, spina bifida, and cerebral palsy) have an upward evolution over the years. In the cases of dyslexia and dyscalculia, the volume of searches on Google during the weeks of the celebration of campaigns was consistently low over all the reviewed years.

### 3.3. Impact on Twitter

Finally, the awareness days that had the greatest impact on Twitter also coincided with the same patterns in the press and on Google: rare diseases, Down syndrome, autism, and disability. With regard to the amount of TT generated, the Down syndrome campaign, with 41 hashtags, stands out among the trends throughout the awareness days for each of the six years (that represents an average of 6.8 TT during the day of the campaign each year). Three campaigns generated more than 20 hashtags over the period from 2015 to 2020; this was the case with rare diseases, autism, and disability (which presented between 24 and 28 TT during the analyzed period). In a lower step but with a considerable presence, the cases of Asperger’s syndrome, cerebral palsy, and spina bifida, with 14, 12, and 8 TT, respectively, also had a wide reach. In contrast, the ADHD, hearing impairment and visual impairment campaigns reached between 4 and 5 TT over the 6 years analyzed in this study, which means that not all years had enough impact on Twitter to reach the consideration of TT.

Awareness campaign days for dyslexia and dyscalculia have, as in the press and on Google, little impact on Twitter. We found two hashtags regarding dyslexia that reached relevant positions in the TT lists, although on different dates (in 2019, the campaign was held on the 10 October, while in 2020, it was the 20 October). For the dyscalculia campaign, TT was not found in any of the years analyzed. In this sense, the TT that reached the best positions are those related to Asperger’s, rare diseases, Down syndrome, autism, cerebral palsy, and disability, all of which were in the top 10 hashtags in the list of trends. Regarding the campaigns that have been maintained for a longer TT, we detail four diagnoses that have been maintained for over more than 85 h on Twitter (rare diseases, Down syndrome, autism, and disability campaigns), followed by Asperger’s syndrome day, which had 65.15 h between trends, and cerebral palsy, with 56 h. There were hashtags that remained for less than 23 h on the trend list. These are the hearing impairment, visual impairment, ADHD, and spina bifida campaigns.

## 4. Discussion

In this paper, we analyzed the effectiveness of world day awareness campaigns for twelve kinds of functional diversity through the study of their impact on the number of news items generated in the press, on the values of searches on Google, and on the TT generated in Twitter.

An analysis of the results showed that the success achieved by the twelve campaigns was uneven. In the case of the campaigns related to disability, rare diseases, Down syndrome, and autism, the levels of impact in the three databases were consistently high in the period from 2015 to 2020, a result that represents progress towards the inclusion of people with functional diversity [6]. During the days of celebration of awareness, campaigns of these diagnoses occurred in more than 50 national press pieces of news; the highest annual Google search volume and an average of 4 and 6.8 TT for each year for each of the days studied in the analyzed period.

Campaigns for the visibility of Asperger’s syndrome, cerebral palsy, and spina bifida achieved medium levels of impact in the analyzed databases. Disability hearing impairment, visual impairment, and ADHD campaigns had lower impact levels in some of the analyzed media. Finally, the dyslexia and dyscalculia campaigns had virtually no impact that was identifiable in either the press or in Google searches, while there were minimal hashtags of interest on Twitter.

The analysis of these results leads us to at least four conclusions. First, the results show that the impact of the campaigns demonstrates comparable results in magnitude in the print media and in the behavior of Internet users (evaluated by the number of searches in Google and the amount and duration of TT on Twitter). Thus, when a campaign reaches high scores in one of these platforms, its success on the other two is often similar.

The correlational nature of the study data does not allow for establishing of causal relationships to determine whether it is the press that generates the increase in Google searches or higher activity levels on Twitter; we also cannot determine whether the increase in Twitter conversations is a factor that generates an increase in other media, or even whether there is some factor outside these elements that is the one that actually determines the other variables.

According to the theory of agenda setting, it is reasonable to suggest that the impact of the campaigns on the press (and probably also on the radio and television, although their data have not been included in the current study) has helped to focus on these issues [11], which are quite possibly linked to an increase in searches on Google and an increase in participation on Twitter. Likewise, some previous studies [27] conclude that topics of conversation on Twitter predict the subsequent web searches of Internet users, which would coincide with the results obtained in this study.

Second, we believe that the differences in the achieved success for each campaign can be explained, at least in part, by the characteristics of the diagnoses themselves. In this sense, campaigns with higher levels of impact correspond to those of diagnoses with greater behavioral problems associated with adaptive or accessibility problems, such as the cases of Down syndrome, rare diseases, or autism. It is possible that both the increase in their prevalence (as happens with autism) and increased awareness by the public (as is the case with rare diseases) interest users and readers of these media. On the contrary, diagnoses that do not have such a severe impact on adaptive functioning, such as ADHD, dyslexia, or dyscalculia, do not seem to be able to attract the spotlight in such a striking way. Visual impairment or hearing impairment also have less impact, perhaps because they are treated as diagnoses that have long been known, which widely justifies the reduced need for awareness campaigns.

The third conclusion of the study is related to the media in which the analyzed campaigns reached higher levels of impact. Usually, we observe more evidence of impact on Twitter and Google than in the press. We believe that this greater success on Google and Twitter may be due to two reasons. On the one hand, it is possible that the actors responsible for the campaigns consider them a priority target because tools such as Twitter are considered very useful social platforms for the dissemination of awareness campaigns, the rapprochement of diagnoses to society, and its influence on users [4,6,19,20,27]. Nonetheless, we also consider it possible that lower presence in the press compared to digital media in the present study could be due to the real possibilities that those responsible for the campaigns advertise through other media outlets such as the press, the radio, or television. This decision might be due to the multitude of interests and variables that come into play in these media.

Finally, as a conclusion, the overall decrease in all media in most of the campaigns during 2020 is possibly due to the exceptional circumstances arising from the COVID-19 pandemic. It is difficult for campaigns to compete with the magnet for attention that constitutes COVID-19. Moreover, [28] concludes that more than three quarters of citizens have changed their information consumption because of the current health emergency. This increase in the consumption of information related to COVID-19 is growing in parallel with media attention and coverage information on the pandemic [28,29,30]. Therefore, it seems reasonable to think that these circumstances are related to the drop in positive trends that recently presented an impact on the analyzed campaigns in this study.

The results obtained in this study allow us some practical implications. We believe that the joint analysis of these results, which show the chronological evolution in the last years of the impact of these 12 campaigns on the press, Google, and Twitter, can be useful information for the planning of future strategies regarding the visibility of such awareness campaigns or projects. Similarly, these results may also be of interest to education, social, and health professionals who also want to increase the levels of social inclusion among people with functional diversity.

Among the limitations of this study, we highlight the complexity to delimit some diagnoses whose nomenclature is imprecise and difficult to specify; thus, the results on Google Trends and in the press may have been partially affected. For example, regarding Campaña Mundial de la Vista, different terms were used in MyNews: “blindness, visual impairment, vision and sight”, leaving on the side labels such as “blind, low vision, vision problems, vision difficulties and low visibility”. The nomenclature changes, derived from the growing concern in recent years for the care of language, added to the great heterogeneity in the subtypes of the same diagnosis. As such, there is a wide disparity in terms, and some campaigns are difficult to narrow down. Another similar limitation, but this time related to Twitter, arises when the hashtags generated for campaign do not contain the name of a specific diagnosis. This fact makes it difficult, and even impossible on some occasions, to identify the location of the hashtags generated for the campaign on the list of trends in this social network. Finally, in this work, we have included only some of the diagnoses with the highest prevalence and presence in the educational environment, excluding other types of functional diversity whose campaigns would also be interesting to analyze.

In this research, we have limited ourselves to exploring the impact of campaigns in three media. Therefore, we have used quantitative data. It would be advisable for future research to carry out a more exhaustive analysis using other media, such as radio and television, and other social networks beyond the press, Twitter, and Google. In this sense, it would be interesting to analyze how functional diversity is perceived in most social networks used by the younger population. In addition, it would be convenient to analyze whether there are awareness campaigns aimed at specific sectors of the population, such as health personnel, social services personnel, or teachers, among other possible sectors that work with people with functional diversity. Regarding the type of analysis carried out, it would be advisable for future studies to make use of qualitative methods so that they could analyze whether these ad hoc increases obtained have an effective influence on increased visibility and awareness of these diagnoses; if they favor legislative changes or increased research; whether the treatment received by the campaigns is sensationalist, paternalistic or, on the contrary, close to an inclusive perspective, etc. In addition, it would be interesting to analyze the content of the pieces of news published in each case, since the fact that there are more pieces of news published on a topic spreads its visibility; however, greater visibility does not necessarily create knowledge about the different diagnoses or forms of social inclusion among these groups, nor does it educate citizens about functional diversity. It is possible that some pieces of news also include the spread of certain myths or misconceptions, so it would be important for future research to analyze this aspect. Finally, we consider that it also would be positive to identify campaigns that have an outstanding impact and analyze what specific factors lead to the success of a campaign in terms of presence in the media, social impact, generation of debate, and effective change in the attitudes in the general population.

## 5. Conclusions

Awareness campaigns, through media, newspapers, and social networks, no matter whether they found in a paper-based or digital media form, help citizens to construct an image about functional diversity. This image could be positive or negative, depending on the content concerning some diagnoses. It is important to obtain the dissemination of these campaigns using different media and social platforms that reach different types of audiences in order to promote a realistic and an inclusive view of functional diversity. These allow for instant dissemination, although it should be considered that any item that may contain false myths or misconceptions can be rapidly shared. In this sense, more research taking into account the importance of the role of various media in addressing issues in society is needed.

## Figures and Tables

**Table 1 ijerph-18-07789-t001:** Types of functional diversity analyzed in this study ^1^.

Type	Specifications	%
	Autism Spectrum Disorder (ASD)	4
Autism	Persistent difficulties in social communication and interaction, restrictive and repetitive patterns of behaviors, interests, or activities	-
Asperger’s syndrome	It is considered today as autism spectrum disorder in a level 1, because its symptoms do not require excessive support	-
	Other neurodevelopmental disorders	6.9
ADHD	Condition that is associated with inattention and/or hyperactivity-impulsivity	-
Dyslexia	Specific learning disorder with a pattern of reading processing difficulties	-
Dyscalculia	Specific learning disorder with difficulties in mathematical processing	-
	Diseases with low prevalence	1.4
Rare diseases	Set of diagnoses with different symptomatology that come together under the same term by sharing certain characteristics: the low prevalence, its chronic character, the required and necessary care, and the continued and expensive treatments. There are around 7000 different rare diseases around the world	-
	Intellectual disability	13.1
Down syndrome	It is the main genetic-chromosomal disorder that may be associated with intellectual disability. It consists of an alteration in chromosome 21, manifested in different degrees of intellectual disability, with recognizable facial features, malformations (usually cardiac and/or digestive), and some risk of epilepsy, leukemia, early aging, or Alzheimer’s, among others	-
	Physical or motor disability	37.3
Spina bifida	Congenital malformation in which the neural tube does not close completely, producing a medullary cyst at the site of the lesion, having permanent sequelae throughout life	-
Cerebral palsy	Disorder of movement and posture development that causes physical and cognitive functional diversity at an early age	5.7
	Sensory disabilities	55.2
Visual impairment	Total or very serious limitation of the visual function	40.9
Hearing impairment	It involves hearing loss of more than 40 dB in the ear with better hearing in adults. In the case of children, it should be more than 30 dB	11.4

^1^ The diagnoses are grouped by the types established in the report of [17]. Only the types and subtypes of functional diversity whose campaigns are analyzed in this research have been included, leaving aside all those diagnoses not covered in the current paper. The data on the percentage that each type of functional diversity represents, with respect to the group of people with functional diversity in Spain (%), come from the same report, made with a representative Spanish sample.

**Table 2 ijerph-18-07789-t002:** Set date in Spain and keywords used in each tool for each of the analyzed campaigns ^1^.

Diagnoses	Awareness Campaign	Date	Keywords on MyNews	Keywords on Google Trends
Asperger’s syndrome	Día Internacional del Síndrome de Asperger	18 February	Asperger	Asperger
Rare diseases	Día Mundial de las Enfermedades Raras	Last day of February	Enfermedad rara, enfermedades raras	enfermedades raras
Down syndrome	Día Mundial del Síndrome de Down	21 March	síndrome de Down	síndrome de Down
Autism	Día Mundial de Concienciación sobre el Autismo	2 April	autism, autista, TEA, Asperger	autismo
Hearing impairment	Día Internacional de las Personas Sordas	Last Saturday on September	sordos, sordera, hipoacusia, discapacidad auditiva, sordas	personas sordas
Cerebral palsy	Día Mundial de la Parálisis Cerebral	First Wednesday of October	parálisis cerebral	parálisis cerebral
Visual impairment	Día Mundial de la Vista	Second Thursday of October	ceguera, discapacidad visual, visión, vista	visión
Dyslexia	Día Internacional de la Dislexia	8 October	dislexia	dislexia
ADHD	Día Nacional del TDAH	27 October	TDAH, trastorno, hiperactividad	TDAH
Spina bifida	Día Internacional de la Espina Bífida	21 November	espina bífida	espina bífida
Disability	Día Internacional de las Personas con Discapacidad	3 December	discapacidad	discapacidad
Dyscalculia	-	w.d.	discalculia	discalculia

^1^ The campaigns are settled in order of celebration’s date. “w.d.” means without a specific date found. The keywords are written down in Spanish.

**Table 3 ijerph-18-07789-t003:** Relationship among the indicators for the period from 2015 to 2020 ^1^.

Campaign	Press	GT	TT	HH:MM
Día Internacional del Síndrome de Asperger	50	78.2	14	66:13
Día Mundial de las Enfermedades Raras	208	100	28	95:10
Día Mundial del Síndrome de Down	205	100	41	155:15
Día Mundial de Concienciación sobre el Autismo	242	100	24	106:10
Día Internacional de las Personas Sordas	20	90.3	5	12:25
Día Mundial de la Vista	22	65.5	5	21:15
Día Mundial de la Parálisis Cerebral	32	93.8	11	56
Día Nacional del TDAH	13	75.6	4	9:25
Día Internacional de la Espina Bífida	12	73.8	8	15:55
Día Internacional de las Personas con Discapacidad	278	99.8	25	87:45

^1^ The campaigns of the International Day of Dyslexia and Dyscalculia are excluded for contradictory information and a lack of consensus on the day of their celebration. Press = total number of pieces of news for the campaign from 2015 to 2020; GT = mean value obtained on Google Trends during the period from 2015 to 2020; TT = total number of trending topics generated on Twitter from 2015 to 2020; HH:MM = total of hours and minutes that the TT was kept on the list of Twitter trends during the studied period.

**Table 4 ijerph-18-07789-t004:** Campaigns with high impact ^1^.

Diagnosis (Celebration’s Day)	Year	Press	GT	TT	HH:MM
Rare diseases (Last day of 28 or 29 February)	2015	37	100	1	00:30
	2016	44	100	4	13:10
	2017	28	100	4	13:25
	2018	40	100	3	12:10
	2019	32	100	10	40:00
	2020	27	100	6	15:55
Down syndrome (21 March)	2015	25	100	1	11:00
	2016	37	100	5	07:20
	2017	34	100	4	14:00
	2018	40	100	5	17:55
	2019	52	100	16	86:35
	2020	17	100	10	18:25
Autism (2 April)	2015	22	100	1	09:10
	2016	45	100	3	15:25
	2017	36	100	2	14:15
	2018	47	100	3	16:50
	2019	43	100	9	35:30
	2020	49	100	6	15:00
Disability (3 December)	2015	47	100	5	13:55
	2016	29	100	1	10:20
	2017	34	99	1	15:10
	2018	62	100	6	16:15
	2019	49	100	8	14:45
	2020	57	100	4	17:20

^1^ GT= value obtained on Google Trends; TT = number of trending topics generated on Twitter; HH:MM = hours and minutes that TT remained on the list of trends on Twitter.

**Table 5 ijerph-18-07789-t005:** Campaigns with medium impact ^1^.

Diagnosis (Celebration’s Day)	Year	Press	GT	TT	HH:MM
Asperger’s syndrome (18 February)	2015	6	100	-	-
	2016	7	100	2	08:05
	2017	3	41	1	10:25
	2018	7	100	3	08:50
	2019	11	28	4	27:18
	2020	16	100	4	11:35
Cerebral palsy (First Wednesday of October)	2015 (07-10)	2	92	1	00:25
	2016 (05-10)	8	71	3	03:10
	2017 (04-10)	8	100	1	09:10
	2018 (03-10)	4	100	2	11:30
	2019 (02-10)	8	100	1	17:05
	2020 (07-10)	2	100	3	14:40
Spina bifida (21 November)	2015	1	100	-	-
	2016	1	80	2	07:05
	2017	2	31	3	05:55
	2018	4	32	1	02:05
	2019	2	100	2	00:50
	2020	2	100	-	-

^1^ The boxes for which no data have been found are marked with “-”. Year: in some cases, not only has the year been specified but so has the specific day; this is because some campaigns suffer variations in the date of celebration regarding the awareness day; GT= value obtained on Google Trends; TT= number of trending topics generated on Twitter; HH:MM= hours and minutes that TT remained on Twitter’s trend list.

**Table 6 ijerph-18-07789-t006:** Campaigns with low impact ^1^.

Diagnosis (Celebration’s Day)	Year	Press	GT	TT	HH:MM
Hearing impairment (Last Saturdayof September)	2015 (28-09)	3	76	-	-
	2016 (22-09)	2	76	1	04:10
	2017 (30-09)	4	100	1	02:50
	2018 (29-09)	3	100	1	00:45
	2019 (28-09)	5	100	2	04:40
	2020 (26-09)	3	90	-	-
Visual impairment (Second Thursday of October)	2015 (08-10)	2	55	-	-
	2016 (13-10)	3	62	1	06:30
	2017 (12-10)	1	86	1	03:50
	2018 (11-10)	6	48	1	03:35
	2019 (10-10)	6	58	1	04:05
	2020 (08-10)	4	84	1	03:15
ADHD (27 October, since 2018)	2015	-	85	-	-
	2016	-	28		-
	2017	-	59	-	-
	2018 (26-10)	6	94	2	07:45
	2019	4	88	-	-
	2020	3	100	2	01:40

^1^ The boxes for which no data have been found are marked with “-”. Year: in some cases, not only has the year been specified but so has the specific day; this is because some campaigns suffer variations in the date of celebration regarding the awareness day; GT = value obtained on Google Trends; TT = number of Trending topics generated on Twitter; HH:MM = hours and minutes that TT remained on Twitter’s trend list.

## Data Availability

All the research data was obtained from the Google Trends, Trendinalia and MyNews databases. No new data were created or analyzed in this study. Data sharing is not applicable to this article.

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
