# Peer review of "Today Is My Day: Analysis of the Awareness Campaigns’ Impact on Functional Diversity in the Press, on Google, and on Twitter"

_ijerph, 2021, doi:10.3390/ijerph18157789_

Round 1
Reviewer 1 Report
In overall, the study shows insight in how to use communication means for a better exposure and for the next step, a better inclusion and understanding of the concept of functional diversity. However, there are some concerns about the text and therefore some recommendations to authors, that may clarify the main points. One major concern is the description of the data received from the three tools. Twitter and the newspapers (digital or paper-based) are clearly described, but the Google search does not really describe what content that has been found. Web pages from the organizations behind the "day", online versions of newspapers or other news media etc.? When referring to the three media bases, it becomes unclear because there is also a mix on the different platforms between professional, personal and organizational content. For example, Twitter includes all kinds: newspapers, organizations and private initiatives. The article would gain in scientific strengtht if there is a stronger description and arguments for what these different media forms found when using Google Trends, Trendinalia and MyNews are used. Also, the discussion would be stronger and not mainly concern the difference between the reach between traditional paper-based media and digital media, but a complex discussion of the different communication means to be used for a better exposure of functional diversity. The traditional theoretical perspective of Agenda Setting will then also add to a more complex but also a more understanding framework of media's role of addressing issues in society (no matter if found in a paper-based or digital media form).
Below are some comments in detail:
19: Check the keywords. There is a discussion in the introduction about the problematic of still using the word disability. Why is then this word a keyword, and why is not functional diversity found here.
43-44: ignorance is used twice in the recital of what the world days fight against.
56: Misspelling of Greta Thunberg's last name.
191-260: Table 3 shows the main results from the study (however, the authors might consider to be more consistent in labelling: Press, Google and Twitter, or MyNews, GT and TT), but to make it easier for readers, follow the logic of the table in the following text: Start with the press and highlight the campaigns with most articles, and then follow with the next from the table and so on. Or change the table to show the Google results firstly and the Press lastly.
261-275: Consider to make this into one table instead to make it more clear for the reader. Or add some commenting text in between the tables to guide the reader of what is most essential in the table.
282-283: As already mentioned, "the three media bases" a bit unclear.
Author Response
We attach a file with the summary of the modifications made.
Reviewer 2 Report
The presented article presents interesting data and conclusions regarding the effectiveness of regional (national) social campaigns using traditional media and social media. The article presents the issues of measuring the effectiveness of this type of campaign in an interesting way. I believe that the article may be published substantially unchanged, however, I would suggest authors to consider the following suggestions. The summary should better highlight the research problem that the manuscript touches on. The description of the methodology does not explain why the authors chose to research the campaigns on 2 social media (Facebook and Twitter), while others were omitted. The key to selecting the campaigns accepted for analysis was also not described.
I would also suggest extending the conclusions to the directions of further possible research, developing the project carried out by the authors.
Author Response

(The authors gave the same response as above.)
